# Awareness of Antibiotics and Antibiotic Resistance in a Rural District of Ha Nam Province, Vietnam: A Cross-Sectional Survey

**DOI:** 10.3390/antibiotics11121751

**Published:** 2022-12-04

**Authors:** Godwin Ulaya, Tu Cam Thi Nguyen, Bich Ngoc Thi Vu, Duc Anh Dang, Hien Anh Thi Nguyen, Hoang Huy Tran, Huong Kieu Thi Tran, Matthew Reeve, Quynh Dieu Pham, Tung Son Trinh, H. Rogier van Doorn, Sonia Lewycka

**Affiliations:** 1Oxford University Clinical Research Unit, Wellcome Africa Asia Programme, National Hospital of Tropical Diseases, Ha Noi 100000, Vietnam; 2Nossal Institute for Global Health, School of Population and Global Health, University of Melbourne, Melbourne, VIC 3004, Australia; 3National Institute for Hygiene and Epidemiology, Ha Noi 100000, Vietnam; 4Centre for Tropical Medicine and Global Health, Nuffield Department of Medicine, University of Oxford, Oxford OX1 2JD, UK

**Keywords:** antibiotic, antibiotic resistance, awareness, knowledge, media, health information, Vietnam

## Abstract

Low awareness of antibiotics and antibiotic resistance may lead to inappropriate antibiotic use and contribute to the problem of antibiotic resistance. This study explored levels and determinants of antibiotic awareness in a rural community in northern Vietnam, through a cross-sectional survey of 324 households in one commune of Ha Nam Province. Awareness and knowledge of antibiotics and antibiotic resistance and determinants were evaluated using structured questionnaires. Most respondents (232/323 (71.8%)) had heard of antibiotics, but fewer could name any antibiotic (68/323 (21.1%)) or had heard of antibiotic resistance (57/322 (17.7%)). In adjusted regression models, antibiotic awareness was lower among those who lived further from health facilities (Odds Ratio (OR): 0.08; 95% Confidence Interval (CI): 0.04–0.19) but higher among those who used interpersonal sources for health information (OR: 4.06; 95% CI: 1.32–12.46). Antibiotic resistance awareness was lower among those who used private providers or pharmacies as their usual health facility (OR: 0.14; 95% CI: 0.05–0.44) but higher among those with medical insurance (OR: 3.70; 95% CI: 1.06–12.96) and those with high media use frequency (OR: 9.54; 95% CI: 2.39–38.07). Awareness of Antimicrobial Resistance (AMR) was also higher among those who sought health information from official sources (OR: 3.88; 95% CI: 1.01–14.86) or had overall high levels of health information seeking (OR: 12.85; 95% CI: 1.63–101.1). In conclusion, communication interventions need to target frequently used media platforms, such as television, as well as key health information providers, such as health workers, as channels for increasing knowledge and changing community antibiotic use behaviour.

## 1. Introduction

The discovery of antibiotics a century ago, followed by a golden age of antibiotic discovery between the 1940s and 1960s, revolutionised medical care; however, the evolution of resistance now threatens modern medicine [1,2]. Antibiotics are instrumental in treating primary bacterial infections as well as reducing infections that occur due to procedures such as surgery, kidney dialysis, organ transplantation, and cancer treatment. As bacterial resistance to antibiotics increases, community-acquired infections and particularly infections following routine medical and surgical procedures are becoming more difficult to treat [3,4]. Antibiotic resistance has been associated with increased morbidity and mortality, a longer hospital stay, and higher hospital costs [5]. An estimated 1.27 million deaths occurred due to antibiotic resistance in 2019 [6]. Without additional interventions, it is estimated that by 2050, there will be 10 million deaths annually due to antibiotic resistance, surpassing projected deaths due to cancer, and [7] the economic impact of antibiotic resistance will be USD100 trillion. Antibiotic resistance disproportionally affects Low- and Middle-Income Countries (LMICs), with higher burdens of infectious disease, poorly regulated antibiotic supply, weak hospital infection prevention and control, and limited access to expensive treatment alternatives and third-line antibiotics [8]. Southeast Asia is considered one of the hotspots for antibiotic resistance selection, emergence, and transmission, and Vietnam has among the highest proportions of resistant pathogens in Asia [6,9].

Consumer practices and behaviours are key factors accelerating antibiotic resistance globally. Non-prescription antibiotic use is high in Vietnam, with 90% of antibiotic sales in pharmacies being made without a prescription; this is driven strongly by consumer demand [10]. Consumer demand for antibiotics is underpinned by limited knowledge of antibiotics and antibiotic resistance. A study conducted in the United States of America (USA) in 1999 found that 27% of the sampled population believed that taking antibiotics when they had a cold would make them recover more quickly, whereas a World Health Organisation (WHO) survey conducted in Vietnam in 2015 indicated that 62% of the respondents believed that antibiotics could cure a cold [11,12].

Interventions that target community antibiotic use awareness have been shown to have a positive impact, although much of this evidence is drawn from high-income countries. For instance, in the USA, community educational campaigns that aimed at raising antibiotic awareness among parents and clinicians resulted in a larger increase in knowledge and awareness regarding appropriate antibiotic use than in control areas [13]. A systematic review conducted in 2017 that included studies from the USA and Europe, recommended that components of multifaceted communication interventions that target both the general public and clinicians could reduce antibiotic use in high-income countries [14].

Few studies on antibiotic awareness have been conducted in LMICs, particularly in rural communities. A 2018 study on antibiotic awareness in five provinces of the central highlands of Vietnam reported that 67.4% of participants had heard of antibiotics, whereas 55.8% had heard of antibiotic resistance [15]. However, adequate knowledge of antibiotics and antibiotic resistance was limited, with only 18.8% being aware that antibiotic use could lead to antibiotic resistance. Another national study conducted in 2019 identified higher education and higher income as being associated with higher antibiotic knowledge; however, this sample was not representative of the general population [16]. A better understanding of community knowledge is needed to inform the development of community-targeted interventions in LMICs, including Vietnam [17].

The main objective of this study was to assess the levels of awareness and knowledge of antibiotics and antibiotic resistance in a rural community in northern Vietnam and to investigate the determinants of awareness and knowledge, in order to inform the development of interventions.

## 2. Results

After stratified random sampling, 389 households were selected for data collection, of which 324 households participated, giving an overall response rate of 83.3% and a margin of error of 5.4% for prevalence estimates. Of those that did not participate, three household respondents refused, whereas the other 62 households had no available respondents found during the study period (Figure 1).

Individual respondent demographic characteristics are summarized in Table 1. There were more female respondents (84.6%) than male respondents. The younger population (18–29 years) were the least represented (16.4%), whereas those above 50 years old were the most represented (43.8%). Most respondents had attended school at some level (81.3%). About two-thirds of respondents (65.8%) were farmers, whereas 28.5% had other types of employment, and the remainder were unemployed.

Access to different media platforms is shown in Figure 2. Television (TV) was the most accessed (97.8%), whereas print media was the least accessed (14.8%).

Health workers (97.2%) and television (95.4%) were the most consulted sources of health information, whereas newspapers (13.3%) and books (15.2%) were the least consulted (Figure 3). Though the commune was small (approximately 9 km^2^ and not more than 5 km at the widest part) and relatively flat, almost half of respondents (48.6%) said they lived more than 10 min travel time to the commune health centre. These respondents were more likely to be low frequency media users (*p* < 0.001) and low health information seeking types (*p* < 0.001).

Nearly three quarters of primary respondents had heard of antibiotics (232/323 (71.8%)), but fewer (68/323 (21.1%)) could name any of the 11 antibiotics on our list spontaneously when asked to give the names of any antibiotics they knew. The most well-known antibiotic was ampicillin/amoxicillin, with only (4.2%) having never heard of it, whereas 98.4%, 94.1%, and 92.3% had never heard of colistin, ciprofloxacin, and Augmentin, respectively, even after probing (Figure 4). About one-fifth (57/322 (17.7%)) of respondents had heard of antibiotic resistance.

In unadjusted models (Table 2), male respondents had lower awareness of antibiotics than female respondents (Odds Ratio (OR): 0.30; 95% Confidence Interval (CI): 0.13–0.73), those who were not working had lower awareness than those who were employed (OR: 0.14; 95% CI: 0.03–0.60), those who used traditional practitioners had lower awareness than those who used government health facilities (OR: 0.23; 95% CI: 0.10–0.57), and those who had to travel more than 10 min to the commune health centre had lower awareness than those who had to travel less than 10 min (OR: 0.06; 95% CI: 0.02–0.15). Those who had attended school had higher awareness than those who had not or whose educational level was unknown (OR: 9.99; 95% CI: 3.90–25.60) (Table 2). After adjusting for age, sex, and education, respondents who travelled more than 10 min to their commune health centre were less likely to be aware of antibiotics than those who travelled less than 10 min (adjusted OR (aOR): 0.08; 95% CI: 0.04–0.19), and those who sought health information from health workers (aOR: 172.78; 95% CI: 13.49–2213.05), the women’s union (aOR: 5.08; 95% CI: 1.71–15.09), and relatives (aOR: 4.08: 95% CI: 1.42–11.74) were more likely to be aware of antibiotics than those who did not. In line with this, respondents who used interpersonal sources of health information were more likely to be aware of antibiotics compared with those with low overall health information seeking behaviour (aOR 4.06, 95% CI 1.32–12.46). Crude and adjusted linear models using the continuous antibiotic knowledge score based on the number of antibiotics recognised, showed qualitatively similar results. Exceptions were that those in the richest household tertile were familiar with more antibiotics than those in poorer households, as were those who used private providers, pharmacies, and drugstores. Those who used television, newspapers, health-workers, and pharmacists as regular health information sources were more familiar with antibiotics than those who did not use these sources. Media use frequency was not associated with antibiotic awareness or knowledge of antibiotics in adjusted models.

Determinants of antibiotic resistance awareness are presented in Table 3. In crude models, older respondents, farmers, and those who were not working were less likely to have heard of antibiotic resistance. After adjusting for age, sex, and education, use of private healthcare, pharmacies, or drugstores was associated with less awareness of antibiotic resistance than the use of government facilities (aOR: 0.14; 95% CI: 0.05–0.44), and in line with this, having a national medical insurance card (to be used at government facilities) was associated with higher awareness of antibiotic resistance (aOR: 3.70; 95% CI: 1.06–12.96). Those with high levels of media use also had higher levels of awareness (aOR:9.54; 95% CI: 2.39–38.07), as did those with high levels of health information seeking (aOR: 12.85; 95% CI: 1.63–101.10), and those that sought health information from official sources (aOR: 3.88; 95% CI: 1.01–14.86) had higher awareness of antibiotic resistance than those with low levels of health information seeking. Specifically, those who sought health information from radio, newspapers, community radio, and social media had higher awareness of antibiotic resistance. In linear regression models based on scores from questions about antibiotic resistance, risk factor patterns were qualitatively similar, with the exception that male respondents had lower scores on antibiotic resistance knowledge and higher household wealth was associated with higher knowledge. A high frequency of media use and higher levels of health information seeking were also associated with higher knowledge, but official health information sources were not associated with higher knowledge. Those who sought health information from television, radio, newspapers, community radio, health workers, the women’s union, relatives, and social media had higher antibiotic resistance knowledge scores.

## 3. Discussion

This study shows that the overall awareness of antibiotics in this rural community in Vietnam is high (71.8%). However, the spontaneous recall or recognition of names of antibiotics from our list was much lower (21.6%). This inconsistency raises the possibility that respondents’ understanding of the word “antibiotic” when reporting awareness was not in line with its actual meaning, as has been observed in other studies. For example, a study in Malaysia in 2010 found that 33% of the respondents confused antibiotics with painkillers such as paracetamol and aspirin [18]. A similar finding was reported in a community-based antibiotic access and use study conducted in six LMICs, including Vietnam [19]. Furthermore, differences between the determinants of antibiotic awareness and antibiotic knowledge scores suggests that having heard of antibiotics in general and more in-depth knowledge in terms of being able to name any of the common antibiotics might be learnt through different sources of information.

This study also established that antibiotic resistance awareness is low (18.2%) in this community and driven by similar demographic and media access factors as antibiotic awareness and knowledge. The level of antibiotic resistance awareness is lower than in a recent study in Vietnam, which reported an awareness of antibiotic resistance of 55.8% in a mixed urban¬–rural population; the difference may be due to our study targeting a rural population [15]. A similar antibiotic awareness study in Thailand conducted between 2017 and 2018 reported a higher antibiotic awareness of up to 95.7% and antibiotic resistance awareness of 74.8%. However, the methodology of measuring this awareness was different; the Thailand study relied on the recognition of pictures of antibiotics [20]. Another possible reason for the high antibiotic awareness in Thailand could be the robust systematic antimicrobial resistance campaigns that preceded the survey from 2012 to 2016 [21]. These studies and their distinct results demonstrate the diversity of approaches to measuring antibiotic awareness and knowledge and the subsequent challenges in comparing different study findings.

Factors shown to be associated with awareness of antibiotics in this study, such as demographic characteristics, type of primary facility, location, and use of media, have been reported before [22,23]. Our study found that those who use private clinics, pharmacies, or drug stores as their usual healthcare provider were familiar with more antibiotics than those who used government facilities, but were less aware of antibiotic resistance and scored lower on the antibiotic resistance knowledge questions. This finding aligns with the commercial orientation of most retail pharmacies and drug stores, particularly in LMICs, where antibiotics can easily be obtained without a prescription. Drug sellers may fail to provide adequate advice about a drug when making a sale, whereas buyers may request antibiotics based on prior experiences or advice from friends or relatives [24]. Thus patients may be more familiar with the names of the products they wish to buy, or that drug sellers wish to sell to them, but have less comprehensive information about the negative effects of these drugs. However, as this was a cross-sectional study, we were unable to establish the direction of causality between awareness and knowledge of antibiotics and antibiotic resistance and healthcare seeking.

In our study, television was the most frequently accessed form of media and also the source used most often for health information. Health workers and community radio were also frequently used sources of health information. We found an association between media use and increased awareness of antibiotics and antibiotic resistance. Other media platforms such as the internet, print media, radio, and SMS are also known to be common sources that individuals use to access information regarding antibiotics. A study in Poland found that the majority of respondents relied on the internet as the source of antibiotic knowledge [25]. Limited research has been performed in LMICs on the role of internet use in raising antibiotic awareness. However, the growing internet use in LMICs, including Vietnam, offers an opportunity for leveraging this platform for raising antibiotic awareness [26]. Our study only explored sources of general health information and which of these media sources were the trusted or primary source of information about antibiotics and antibiotic resistance, and the content of their messages remains to be investigated in this population. Research investigating media representations of AMR in the UK and China suggests that media content needs to be reoriented to communicate actions consumers could take to tackle AMR [27].

Although the study area was small and relatively flat, we found that living more than 10 min of travel time from the usual health facility was associated with lower antibiotic awareness. Further investigation into the relationship between distance from health facility and access to health information revealed that those who lived more than 10 min travel time away were low frequency media users and low health information seeking types. This finding, along with the finding that health workers were important sources of both antibiotic and antibiotic resistance knowledge, highlights the importance of health facilities as sources of health information [28].

This study is among the few studies in LMICs that have evaluated the antibiotic awareness within the general public and particularly rural communities. Moreover, this study incorporated diverse measurement approaches to triangulate antibiotic awareness. The response rate is also relatively high and therefore the risk of selection bias is reduced.

This study has several limitations. The results represent one rural community in northern Vietnam and may not be generalizable to other settings in Vietnam. The cross-sectional design limits any conclusions regarding the direction of causal effects. Furthermore, we cannot rule out the possibility of various information biases. For instance, even though data collectors were trained, the interviewers were nurses from the local health centre and may have interpreted the responses using their health knowledge. Respondents may have had difficulty recalling some information related to health-seeking behaviours, and this might have introduced recall bias. Respondents may also have introduced bias by reporting what they thought were desirable/socially-acceptable behaviours. Antibiotics may be known by a mixture of generic and brand names, and the list of antibiotic names we provided might not have been the most commonly used in the area. We are conducting an ethnographic study to explore understanding of antibiotics and the language used to describe them that will aid interpretation of these results and inform future research in this area. Lastly, there might be other unmeasured factors that influence awareness of antibiotics and antibiotic resistance or confound the observed associations.

## 4. Materials and Methods

### 4.1. Study Design

The study was part of a large quantitative cross-sectional household survey conducted with 324 households in a rural commune of Ha Nam Province, northern Vietnam, between 16 July 2018 and 9 April 2019. The commune had a population of 9746 people. The study design has been described in detail elsewhere [29].

### 4.2. Sample Size

A random sample of households was selected from a household list obtained from the commune health centre. Sampling was stratified so that households with children below five years were oversampled, as one of the study’s primary aims was to collect samples and conduct antimicrobial susceptibility testing for *Streptococcus pneumoniae*, which is carried at higher levels among children under five years old. A sample size of 340 would allow us to estimate prevalence with a 5.2% margin of error with 95% confidence. We sampled 390 households, allowing for a 15% drop-out rate. Nurses from the commune health centre were trained to administer structured questionnaires (in Vietnamese) and to collect information on healthcare seeking behaviour, awareness of antibiotics and antibiotic resistance, and general socio-demographic characteristics. For every household, one adult caregiver responded to the questionnaire. Eligibility depended on giving written informed consent to the study and study procedures, and there were no exclusion criteria for this study.

### 4.3. Definitions of Variables

The definitions of dependent and independent variables are provided in Table 4. Awareness of antibiotics and antibiotic resistance were measured by asking primary respondents whether they had heard of antibiotics or antibiotic resistance. [15] “No” and “don’t know” responses were grouped into one negative response (those who were not confident they had heard of antibiotics) and compared with “yes” responses (those who were confident they had heard of antibiotics) to create a binary variable. The respondents who had heard of antibiotics were asked whether they could spontaneously name any antibiotics, then a list of common antibiotics (including penicillin, doxycycline, tetracycline, erythromycin, streptomycin, Augmentin (amoxicillin + clavulanate), cephalexin, cotrimoxazole, ciprofloxacin, ampicillin/amoxicillin, and colistin) was read out and respondents were asked to say whether they had ever heard of them. We used binary variables for awareness of antibiotics and antibiotic resistance and also generated an antibiotic knowledge score using responses to the spontaneous naming and recognition of antibiotics and an antibiotic resistance knowledge score based on answers to questions about consequences of and ways of preventing antibiotic resistance.

### 4.4. Data Analysis

Data analysis utilized STATA software version 17.0. Variable categories of similar characteristics were re-grouped into fewer categories to reduce data sparsity. Descriptive statistics were performed to determine the distribution of individual respondent characteristics. Point prevalences were derived for dependent variables, awareness of antibiotics and awareness of antibiotic resistance. Bivariable and multivariable logistic regression models were performed to show associations between media use, health information sources, distance to a health facility, usual health facility, and having a medical insurance card and awareness of antibiotics and antibiotic resistance. Variables such as age, sex, education, occupation, and household wealth were evaluated to be potential confounding factors and were adjusted for in the multivariable logistic models. Bivariable and multivariable linear regression models were run with the continuous antibiotic knowledge and antibiotic resistance scores.

## 5. Conclusions

Self-reported awareness of antibiotics was high in this study, though knowledge about types of antibiotics and awareness of antibiotic resistance was much lower. The determinants of antibiotic and antibiotic resistance awareness and knowledge included socio-demographic factors (especially age and occupation) and access to healthcare (usual facility type, distance from health facility). Health workers and interpersonal health information sources were associated with more awareness of antibiotics. High use of media (particularly TV, radio, community radio/loudspeaker, newspaper, and social media) was associated with more awareness of AMR. The complex relationships between knowledge, media access, health information, and behaviour, makes identifying appropriate intervention strategies challenging. Targeting multiple media channels and health information sources, particularly those with the highest access and use, such as television and health providers, is likely to have the highest reach for communication campaigns. Multi-modal campaigns may also be beneficial.

## Figures and Tables

**Figure 1 antibiotics-11-01751-f001:**
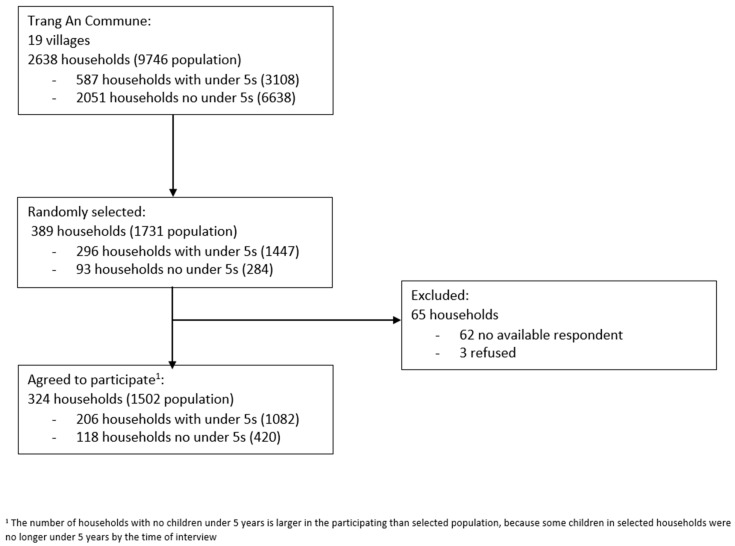
Flow diagram of participant inclusion.

**Figure 2 antibiotics-11-01751-f002:**
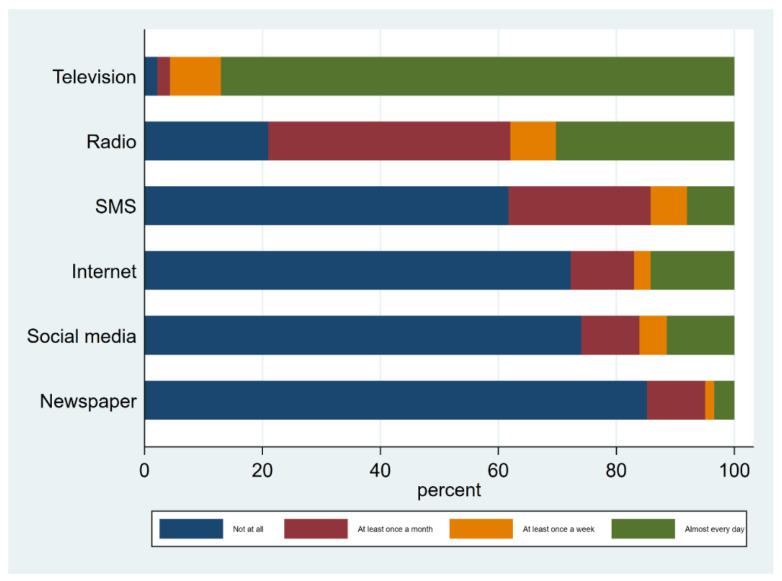
Frequency of media use ((SMS = Short Message Service).

**Figure 3 antibiotics-11-01751-f003:**
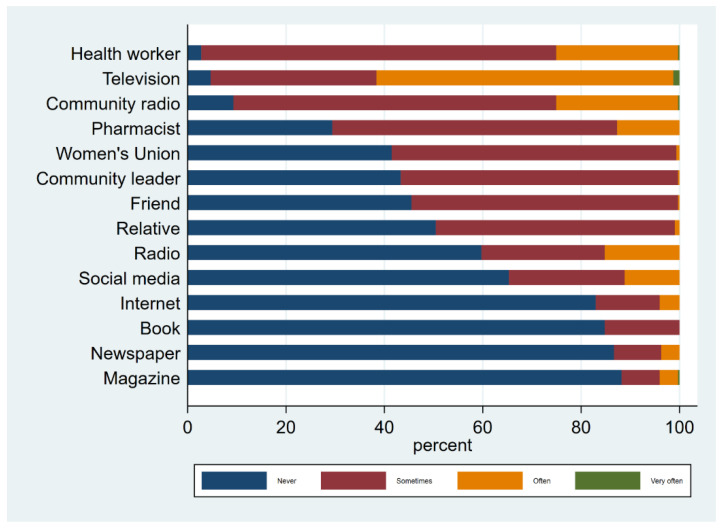
Frequency of access to health information sources.

**Figure 4 antibiotics-11-01751-f004:**
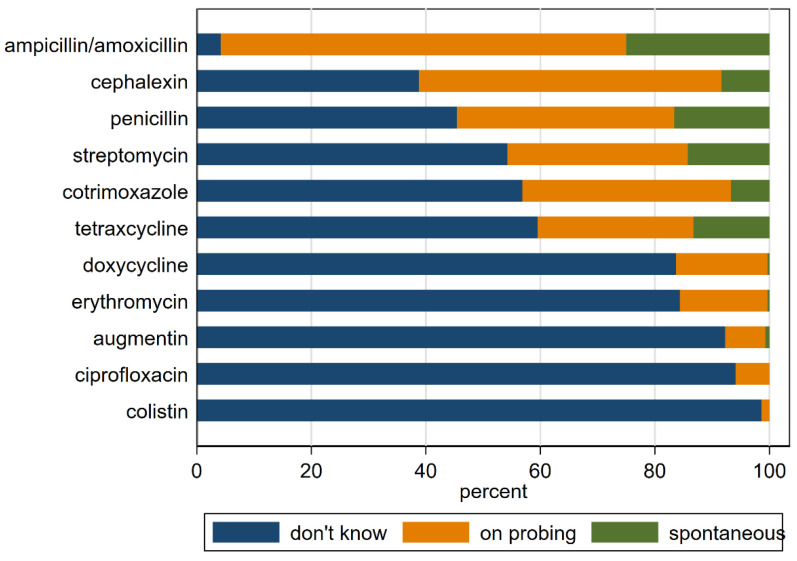
Antibiotic knowledge.

**Table 1 antibiotics-11-01751-t001:** Socio-demographics for the study population.

Characteristic		Total Study Population	
		*n*	%
Age (years)	18–29 years	53	16.4
	30–49 years	129	39.8
	50 years and above	142	43.8
Sex	Female	274	84.6
	Male	50	15.4
Highest level of education	Never attended school/unknown	59	18.7
	Attended any school level	257	81.3
Occupation	Employed	91	28.5
	Farmer	210	65.8
	Not working	18	5.6
Household wealth tertile	Poor	115	35.5
	Middle	104	32.1
	Rich	105	32.4
Used print media in last month (newspapers and magazines)	Not used	276	85.2
	Used	48	14.8
Listened to radio in last month	Not used	68	21
	Used	256	79
Watched television in last month	Not used	7	2.2
	Used	317	97.8
Used Short Message Service (SMS) in last month	Not used	200	61.7
	Used	124	38.3
Used internet in last month	Not used	234	72.2
	Used	90	27.8
Used social media in last month	Not used	240	74.1
	Used	84	25.9
Television	Never	15	4.6
	Sometimes, often, very often	308	95.4
Radio	Never	193	59.8
	Sometimes, often, very often	130	40.2
Newspaper	Never	280	86.7
	Sometimes, often, very often	43	13.3
Magazine	Never	285	88.2
	Sometimes, often, very often	38	11.8
Book	Never	274	84.8
	Sometimes, often, very often	49	15.2
Community radio	Never	30	9.3
	Sometimes, often, very often	293	90.7
Health worker	Never	9	2.8
	Sometimes, often, very often	314	97.2
Pharmacist	Never	95	29.4
	Sometimes, often, very often	228	70.6
Community leader	Never	140	43.3
	Sometimes, often, very often	183	56.7
Women’s Union	Never	134	41.5
	Sometimes, often, very often	189	58.5
Relative	Never	163	50.5
	Sometimes, often, very often	160	49.5
Friend	Never	147	45.5
	Sometimes, often, very often	176	54.5
Internet	Never	268	83
	Sometimes, often, very often	55	17
Social media	Never	211	65.3
	Sometimes, often, very often	112	34.7

**Table 2 antibiotics-11-01751-t002:** Univariable and multivariable regression modelling of antibiotic awareness.

		Total	Heard of Antibiotics		Not Heard of Antibiotics		Heard of Antibiotics						Antibiotic Knowledge Score (Based on Naming Antibiotics)					
							Crude Odds Ratio			Adjusted Odds Ratio			Crude Regression Coefficient			Adjusted Regression Coefficient		
Characteristic		N	*n*	%	*n*	%	OR	95% CI	*p*-Value	aOR	95% CI	*p*-Value	B	95% CI	*p*-Value	aB	95% CI	*p*-Value
Total		323	232	71.8	91	28.2												
Age (years)	18–29 years	52	40	76.9	12	23.1	1											
	30–49 years	127	98	77.2	29	22.8	1.42	(0.44–4.56)	0.556				1.69	(−1.12–4.50)	0.237			
	50 years and above	136	90	66.2	46	33.8	0.42	(0.14–1.26)	0.123				−1.50	(−4.41–1.41)	0.312			
Sex	Female	267	204	76.4	63	23.6	1											
	Male	48	24	50	24	50	0.30	(0.13–0.73)	0.008				−3.16	(−5.66–−0.66)	0.013			
Highest level of education	Never attended school/unknown	59	15	25.4	44	74.6	1											
	Attended any school level	256	213	83.2	43	16.8	9.99	(3.90–25.60)	<0.001				2.26	(0.13–4.39)	0.037			
Occupation	Employed	91	71	78	20	22	1											
	Farmer	203	150	73.9	53	26.1	0.59	(0.26–1.38)	0.225				−1.56	(−3.63–0.50)	0.137			
	Not working	17	5	29.4	12	70.6	0.14	(0.03–0.60)	0.008				−5.78	(−9.69–−1.86)	0.004			
Household wealth tertile	Poor	109	75	68.8	34	31.2	1											
	Middle	102	79	77.5	23	22.5	1.22	(0.54–2.78)	0.626				1.72	(−0.61–4.04)	0.148			
	Rich	104	74	71.2	30	28.8	1.75	(0.78–3.95)	0.177				2.43	(0.11–4.75)	0.040			
Usual health facility	Government facility	218	165	75.7	53	24.3	1											
	Private/pharmacy/drugstore	50	40	80	10	20	1.70	(0.58–4.97)	0.331	1.54	(0.48–4.96)	0.468	3.51	(0.97–6.05)	0.007	4.07	(1.70–6.43)	0.001
	Traditional practitioner	47	23	48.9	24	51.1	0.23	(0.10–0.57)	0.001	0.40	(0.13–1.20)	0.102	−2.70	(−5.28–−0.12)	0.040	−1.29	(−3.80–1.23)	0.315
Distance to nearest health facility	Less than 10 min	151	136	90.1	15	9.9	1											
	10 min or more	160	89	55.6	71	44.4	0.06	(0.02–0.15)	<0.001	0.08	(0.04–0.19)	<0.001	−2.77	(−4.56–−0.98)	0.003	−1.43	(−3.24–0.38)	0.121
Medical insurance card	No	63	41	65.1	22	34.9	1											
	Yes	238	181	76.1	57	23.9	1.11	(0.48–2.57)	0.814	1.77	(0.61–5.13)	0.289	−1.07	(−3.36–1.22)	0.360	−0.57	(−2.66–1.52)	0.592
Media use frequency	Low	120	92	76.7	28	23.3	1											
	Medium	89	53	59.6	36	40.4	0.34	(0.15–0.76)	0.009	0.41	(0.14–1.24)	0.114	−2.13	(−4.63–0.36)	0.093	−2.01	(−4.82–0.79)	0.159
	High	106	83	78.3	23	21.7	1.42	(0.59–3.40)	0.435	1.67	(0.43–6.52)	0.459	0.32	(−1.71–2.35)	0.756	−0.30	(−2.61–2.01)	0.798
Health information source																		
Television	Never	14	4	28.6	10	71.4	1											
	Sometimes, often, very often	301	224	74.4	77	25.6	14.62	(2.81–75.93)	0.001	7.32	(0.44–122.44)	0.166	7.21	(3.73–10.70)	<0.001	5.15	(1.10–9.20)	0.013
Radio	Never	190	154	81.1	36	18.9												
	Sometimes, often, very often	125	74	59.2	51	40.8	0.48	(0.24–0.96)	0.039	0.81	(0.33–1.99)	0.641	−0.22	(−2.15–1.70)	0.820	0.54	(−1.39–2.48)	0.582
Newspaper	Never	273	206	75.5	67	24.5												
	Sometimes, often, very often	42	22	52.4	20	47.6	0.50	(0.19–1.32)	0.159	1.91	(0.51–7.15)	0.333	0.92	(−1.05–2.89)	0.359	2.36	(0.12–4.60)	0.039
Magazine	Never	278	212	76.3	66	23.7												
	Sometimes, often, very often	37	16	43.2	21	56.8	0.31	(0.11–0.88)	0.028	1.17	(0.30–4.53)	0.820	0.46	(−1.74–2.65)	0.683	2.32	(−0.45–5.08)	0.100
Book	Never	268	205	76.5	63	23.5												
	Sometimes, often, very often	47	23	48.9	24	51.1	0.30	(0.12–0.74)	0.010	0.57	(0.21–1.54)	0.266	−0.30	(−2.13–1.53)	0.748	0.38	(−1.69–2.45)	0.721
Community radio	Never	28	17	60.7	11	39.3												
	Sometimes, often, very often	287	211	73.5	76	26.5	3.07	(1.06–8.90)	0.039	2.95	(0.65–13.37)	0.159	4.20	(0.44–7.95)	0.029	2.80	(−0.62–6.21)	0.108
Health worker	Never	8	1	12.5	7	87.5												
	Sometimes, often, very often	307	227	73.9	80	26.1	104.80	(12.14–904.50)	<0.001	172.78	(13.49–2213.05)	<0.001	12.00	(8.60–15.40)	<0.001	11.31	(8.27–14.35)	<0.001
Pharmacist	Never	92	64	69.6	28	30.4												
	Sometimes, often, very often	223	164	73.5	59	26.5	2.23	(1.09–4.59)	0.029	2.45	(0.88–6.87)	0.087	3.38	(1.20–5.56)	0.003	2.67	(0.57–4.78)	0.013
Community leader	Never	137	94	68.6	43	31.4												
	Sometimes, often, very often	178	134	75.3	44	24.7	1.77	(0.90–3.50)	0.099	2.53	(0.99–6.44)	0.052	0.24	(−1.87–2.35)	0.823	−0.33	(−2.39–1.73)	0.754
Woman’s Union	Never	131	84	64.1	47	35.9												
	Sometimes, often, very often	184	144	78.3	40	21.7	2.46	(1.24–4.90)	0.011	5.08	(1.71–15.09)	0.004	2.14	(0.14–4.14)	0.037	1.51	(−0.52–3.55)	0.145
Relative	Never	160	112	70	48	30												
	Sometimes, often, very often	155	116	74.8	39	25.2	1.68	(0.85–3.33)	0.136	4.08	(1.42–11.74)	0.009	−0.59	(−2.44–1.26)	0.529	−0.95	(−2.73–0.84)	0.297
Friend	Never	143	96	67.1	47	32.9												
	Sometimes, often, very often	172	132	76.7	40	23.3	1.56	(0.79–3.09)	0.199	1.73	(0.69–4.34)	0.243	−0.24	(−2.12–1.64)	0.801	−1.51	(−3.30–0.28)	0.098
Internet	Never	262	186	71	76	29												
	Sometimes, often, very often	53	42	79.2	11	20.8	1.91	(0.65–5.59)	0.240	2.95	(1.10–7.94)	0.032	0.77	(−0.72–2.27)	0.310	0.19	(−1.83–2.21)	0.855
Social media	Never	207	147	71	60	29												
	Sometimes, often, very often	108	81	75	27	25	1.63	(0.76–3.48)	0.206	2.62	(0.69–9.99)	0.157	0.47	(−1.47–2.42)	0.631	0.48	(−1.83–2.80)	0.683
Health information seeking type	Low information seeking	124	94	75.8	30	24.2												
	Official sources	34	16	47.1	18	52.9	0.32	(0.11–0.97)	0.044	0.48	(0.13–1.81)	0.277	−3.90	(−6.75–−1.04)	0.008	−2.56	(−5.56–0.43)	0.093
	Interpersonal sources	123	105	85.4	18	14.6	2.05	(0.88–4.80)	0.096	4.06	(1.32–12.46)	0.015	−1.50	(−3.64–0.64)	0.169	−1.63	(−3.67–0.40)	0.116
	High information seeking	34	13	38.2	21	61.8	0.38	(0.12–1.17)	0.092	2.12	(0.37–12.14)	0.398	−0.61	(−3.32–2.10)	0.659	0.44	(−2.56–3.44)	0.772

**Table 3 antibiotics-11-01751-t003:** Univariable and multivariable regression modelling of antibiotic resistance awareness.

		Total	Heard of AMR		Not Heard of AMR		Heard of Antibiotic Resistance (AMR)						Antibiotic Resistance Knowledge Score (Based on Questions about Antibiotic Resistance)					
							Crude Odds Ratio			Adjusted Odds Ratio			Crude Regression Coefficient			Adjusted Regression Coefficient		
Characteristic		N	*n*	%	*n*	%	OR	95% CI	*p*-Value	aOR	95% CI	*p*-Value	B	95% CI	*p*-Value	aB	95% CI	*p*-Value
Total		322	57	17.7	265	82.3												
Age (years)	18–29 years	52	21	40.4	31	59.6	1											
	30–49 years	127	26	20.5	101	79.5	0.40	(0.14–1.15)	0.089				−3.11	(−6.84–0.63)	0.103			
	50 years and above	135	10	7.4	125	92.6	0.09	(0.03–0.33)	<0.001				−6.44	(−9.81–−3.06)	<0.001			
Sex	Female	266	49	18.4	217	81.6	1											
	Male	48	8	16.7	40	83.3	0.49	(0.14–1.68)	0.253				−2.68	(−4.11–−1.25)	<0.001			
Highest level of education	Never attended school/unknown	59	6	10.2	53	89.8	1											
	Attended any school level	255	51	20	204	80	1.20	(0.37–3.89)	0.764				1.44	(−1.05–3.92)	0.256			
Occupation	Employed	90	25	27.8	65	72.2	1											
	Farmer	203	29	14.3	174	85.7	0.33	(0.14–0.78)	0.012				−3.54	(−6.11–−0.97)	0.007			
	Not working	17	3	17.6	14	82.4	0.10	(0.02–0.43)	0.002				−5.40	(−7.94–−2.87)	<0.001			
Household wealth tertile	Poor	109	15	13.8	94	86.2	1											
	Middle	103	14	13.6	89	86.4	0.97	(0.31–3.02)	0.957				−0.03	(−1.99–1.93)	0.974			
	Rich	102	28	27.5	74	72.5	2.60	(0.97–6.96)	0.057				2.66	(0.26–5.06)	0.030			
Usual health facility	Government facility	219	43	19.6	176	80.4	1											
	Private/pharmacy/drugstore	49	5	10.2	44	89.8	0.14	(0.05–0.39)	<0.001	0.14	(0.05–0.44)	0.001	−2.18	(−3.76–−0.61)	0.007	−1.96	(−3.41–−0.51)	0.008
	Traditional practitioner	46	9	19.6	37	80.4	0.91	(0.30–2.78)	0.869	1.55	(0.41–5.82)	0.512	−1.95	(−4.19–0.30)	0.089	−0.56	(−2.85–1.74)	0.635
Distance to nearest health facility	Less than 10 min	151	38	25.2	113	74.8	1											
	10 min or more	160	19	11.9	141	88.1	0.31	(0.13–0.74)	0.009	0.43	(0.18–1.07)	0.068	−4.29	(−6.27–−2.31)	<0.001	−3.26	(−5.20–−1.33)	0.001
Medical insurance card	No	62	8	12.9	54	87.1	1											
	Yes	238	48	20.2	190	79.8	2.94	(0.86–10.03)	0.085	3.70	(1.06–12.96)	0.041	2.42	(0.97–3.86)	0.001	2.85	(1.35–4.36)	<0.001
Media use frequency	Low	121	8	6.6	113	93.4	1											
	Medium	89	7	7.9	82	92.1	1.36	(0.31–5.98)	0.686	1.38	(0.30–6.31)	0.680	0.09	(−1.68–1.87)	0.919	0.69	(−1.35–2.73)	0.506
	High	104	42	40.4	62	59.6	11.60	(3.73–36.09)	<0.001	9.54	(2.39–38.07)	0.001	5.86	(3.44–8.28)	<0.001	5.46	(2.49–8.42)	<0.001
Health information source																		
Television	Never	14	0	0	14	100	1											
	Sometimes, often, very often	299	57	19.1	242	80.9	1.00			1.00			3.98	(2.97–4.98)	<0.001	2.34	(0.65–4.03)	0.007
Radio	Never	189	26	13.8	163	86.2												
	Sometimes, often, very often	124	31	25	93	75	1.89	(0.82–4.34)	0.135	3.10	(1.09–8.81)	0.034	1.12	(−0.89–3.12)	0.273	2.25	(0.20–4.30)	0.032
Newspaper	Never	271	41	15.1	230	84.9												
	Sometimes, often, very often	42	16	38.1	26	61.9	4.53	(1.60–12.83)	0.005	9.37	(1.82–48.15)	0.008	4.34	(0.16–8.52)	0.042	5.82	(1.78–9.86)	0.005
Magazine	Never	276	45	16.3	231	83.7												
	Sometimes, often, very often	37	12	32.4	25	67.6	3.61	(1.17–11.10)	0.025	5.88	(0.85–40.71)	0.072	2.78	(−1.32–6.87)	0.183	3.53	(−0.37–7.43)	0.076
Book	Never	266	42	15.8	224	84.2												
	Sometimes, often, very often	47	15	31.9	32	68.1	2.15	(0.77–6.01)	0.142	1.85	(0.40–8.66)	0.432	1.92	(−1.50–5.34)	0.271	2.06	(−2.11–6.24)	0.332
Community radio	Never	28	5	17.9	23	82.1												
	Sometimes, often, very often	285	52	18.2	233	81.8	3.83	(1.28–11.46)	0.016	3.82	(1.15–12.67)	0.029	3.35	(2.09–4.61)	<0.001	3.23	(1.63–4.84)	<0.001
Health worker	Never	8	1	12.5	7	87.5												
	Sometimes, often, very often	305	56	18.4	249	81.6	8.28	(0.95–72.26)	0.056	5.05	(0.60–42.30)	0.135	3.43	(2.00–4.87)	<0.001	2.53	(0.60–4.47)	0.010
Pharmacist	Never	91	22	24.2	69	75.8												
	Sometimes, often, very often	222	35	15.8	187	84.2	1.10	(0.46–2.66)	0.826	0.91	(0.33–2.56)	0.861	1.22	(−0.60–3.04)	0.189	0.87	(−0.84–2.58)	0.319
Community leader	Never	135	22	16.3	113	83.7												
	Sometimes, often, very often	178	35	19.7	143	80.3	1.29	(0.55–3.02)	0.550	1.29	(0.52–3.19)	0.580	1.36	(−0.50–3.23)	0.151	1.43	(−0.40–3.26)	0.125
Woman’s Union	Never	129	20	15.5	109	84.5												
	Sometimes, often, very often	184	37	20.1	147	79.9	1.70	(0.72–4.05)	0.227	1.43	(0.58–3.56)	0.439	2.95	(1.22–4.68)	0.001	2.77	(1.10–4.44)	0.001
Relative	Never	158	27	17.1	131	82.9												
	Sometimes, often, very often	155	30	19.4	125	80.6	1.54	(0.67–3.52)	0.306	1.46	(0.63–3.40)	0.379	2.06	(0.15–3.96)	0.035	2.09	(0.27–3.91)	0.024
Friend	Never	141	18	12.8	123	87.2												
	Sometimes, often, very often	172	39	22.7	133	77.3	1.93	(0.82–4.53)	0.130	1.20	(0.52–2.77)	0.675	2.40	(0.51–4.29)	0.013	1.37	(−0.41–3.15)	0.132
Internet	Never	260	35	13.5	225	86.5												
	Sometimes, often, very often	53	22	41.5	31	58.5	3.29	(1.26–8.64)	0.016	2.00	(0.74–5.40)	0.173	3.17	(−0.27–6.62)	0.071	1.81	(−1.43–5.05)	0.272
Social media	Never	207	20	9.7	187	90.3												
	Sometimes, often, very often	106	37	34.9	69	65.1	6.16	(2.58–14.74)	<0.001	4.43	(1.65–11.86)	0.003	4.30	(1.88–6.72)	0.001	3.23	(0.64–5.82)	0.015
Health information seeking type	Low information seeking	124	20	16.1	104	83.9												
	Official sources	32	6	18.8	26	81.3	1.92	(0.46–8.06)	0.369	3.88	(1.01–14.86)	0.048	0.28	(−2.60–3.15)	0.849	1.50	(−0.94–3.95)	0.227
	Interpersonal sources	123	21	17.1	102	82.9	1.25	(0.47–3.34)	0.654	1.33	(0.48–3.66)	0.583	1.53	(−0.40–3.46)	0.119	1.74	(−0.13–3.60)	0.067
	High information seeking	35	10	28.6	25	71.4	5.08	(1.48–17.51)	0.010	12.85	(1.63–101.10)	0.015	4.77	(0.03–9.51)	0.048	6.75	(1.88–11.62)	0.007

**Table 4 antibiotics-11-01751-t004:** Definitions of independent variables and their categories.

Variable	Definition	Categories
Antibiotic awareness	Have you ever heard of a type of medicine called an antibiotic?	Yes No/don’t know
Antibiotic knowledge	Which antibiotics have you heard of? (Mentioned spontaneously or after probing—penicillin, doxycycline, tetracycline, erythromycin, ampicillin/amoxicillin, Augmentin, streptomycin, cotrimoxazole, cephalexin, ciprofloxacin, colistin)	Score (sum of scores for each antibiotic, where 3 for each antibiotic mentioned spontaneously, 2 for each antibiotic mentioned after probing, and 1 for each antibiotic not known)
Antibiotic resistance awareness	Some antibiotic medicines that used to work in fighting infections no longer work. This problem is called antibiotic resistance. Have you heard of this problem before?	Yes No/don’t know
Antibiotic resistance knowledge	What could the consequences of getting an antibiotic resistant infection be? (Multiple choice answers: Be sick for longer; May have to visit doctor more or be treated in hospital; May need more expensive medicine that may cause side-effects; Other) Can you think of any ways of reducing the problem of antibiotic resistance? (Don’t take antibiotics when they are not needed (e.g., for colds and sore throats); Don’t demand antibiotics from health-workers or drug suppliers; Make sure antibiotics are good quality and within expiry date; Complete the course as recommended by a health-worker; Don’t use antibiotics prescribed for someone else; Make sure you use the right antibiotic for the right infection; Make sure you take antibiotics as soon as you feel sick; Make sure you take a very strong antibiotic to kill the infection; Take several different antibiotics to make sure the infection is killed; Don’t use antibiotics in animal feed as a growth promoter; Washing hands after contact with a live animal, slaughtering animals, or preparing meat; Washing hands after contact with someone or something that has been touched by a person who has an antibiotic-resistant infection)	Score summing each correct answer
Education	Whether one had attended any school beyond nursery level	Any school level—primary to tertiary Never attended school—less than primary school level—or unknown
Occupation	Main work of the respondent	Farmers Employed (factories, labourers, office, shops and others) Not working
Household wealth	Tertiles based on principal component analysis of household assets (telephone, mobile phone, computer, tablet, radio, TV, bed, table and chairs, sofa, fan, air conditioner, gas cooker, electric cooker, washing machine, refrigerator, bicycle, motorcycle car, tractor, motorboat), electricity, crowding, type of flooring, type of roofing, type of walls	Poor Middle Rich
Usual health facility	The facility that respondents considered as their primary facility where their children or themselves go when they get sick	Government Private/pharmacy/drug stores Others (traditional healers, shops, and those that do not seek care)
Distance to health facility	Time it took from their house to the commune primary healthcare centre	Less than 10 min More than 10 min
Medical insurance card	Whether they had a government provided medical insurance card	Yes—when they had the card No—when they did not have
Access to different media	Respondents were asked whether they had access to different media platforms	Access to Print media, Radio, Television, SMS, social media and Internet No access to the above.
Media use frequency	Tertiles based on principal component analysis of frequency of access to different media platforms	Low Medium High
Health information seeking type	Groups based on latent class analysis of frequency of access to different sources of health information	Low information seeking across all sources Official sources-mainly newspaper, television, radio, community radio, health worker Interpersonal sources-mainly pharmacist, friends, relatives, community leader, women’s union High information seeking across all sources

## Data Availability

The datasets generated and analysed during the current study are not publicly available as this was not included in the consent process, but anonymised datasets can be made available from the corresponding author on reasonable request.

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
