# Peer review of "Awareness of Antibiotics and Antibiotic Resistance in a Rural District of Ha Nam Province, Vietnam: A Cross-Sectional Survey"

_antibiotics, 2022, doi:10.3390/antibiotics11121751_

Round 1

Reviewer 1 Report

The manuscript by Dr. Ulaya and coworkers used survey data to assess the levels and determinants of antibiotic awareness in a rural community in northern Vietnamese. This is an interesting and important topic, but there are several major and minor flaws in this study.

Major:

1. In the Materials and Methods section, (1) page 14, line 316: the authors need to carefully state the survey period (e.q. ….between January 1, 2018 and March 31, 2019), not only the year. (2) respondents answered “don’t know” may imply that one lack knowledge or interest in the questions about awareness and knowledge of antibiotics and antibiotic resistance asked in the survey. In addition, it cannot be determined whether “don’t know” responses reflected an uncertainty which information had been seen or a desire to minimize response time. Please explain and add more information to a discussion. (3) Why did the authors group ‘no’ and ‘don't know’ together in the bivariable and multivariable regression models? Please explain and clarify more detail in the text.

2. In the Results section, (1) the authors have to show the aOR/aB, 95%CI and p-value of covariates for age, sex, highest level of education, occupation and household wealth tertile in multivariable logistic and linear regression models in Table 2 and Table 3. (2) according to the results of Table 2 and Table 3, why did the authors not present the aB values of significant variables in the sentences? Please clarify more detail in the text.

3. In the Discussion section, (1) page 14, line 306‒310: a certain degree of recall bias is an inherent limitation to the study that uses self-reported survey data. In terms of attitudes and experiences of the household respondents, the varied reasons for awareness and knowledge of common antibiotics and antibiotic resistance and healthcare seeking, this could lead to the recall bias. (2) other unmeasured factors, such as access to medical care, medication use, and the average number of medical outpatient visits in the past month may also influence the awareness and knowledge of antibiotics and healthcare seeking among the household respondents. (3) these results may generalize only to a rural community in northern Vietnamese because of diverse household’s practice patterns. The results are less generalizable to other rural communities, provinces or countries. Please make clear about the limitation of the study in the text.

4. In the Conclusion section, the description is too general and simplistic (i.e., knowledge about types of antibiotics, awareness of antibiotic resistance, and determinants of antibiotic and antibiotic resistance awareness). These significant differences of main findings need to be summarized and rewritten. Please clarify and modify this more detail in the text.

5. The ‘citation formatting’ of several references (Ref. 1, 4, 7, 8, 12, 13, 15, 16, 18, 19, 22, 27, 29) is inaccuracy and inconsistent, for example the article title, journal name, page number, etc.. Please check reference formatting in detail according to the instructions of the Information for Authors of Antibiotics.

Minor:

1. Abbreviations (i.e., AMR, WHO, ….) should be defined in full words (i.e., antimicrobial resistance (AMR), World Health Organization (WHO), ….) in the first time of use, and then the abbreviation can be use many times.

2. Page 2, lines 65: “…have a positive impact, although much of this evidence…” Please delete the comma after “impact”.

3. The abbreviation (i.e. SMS) should be defined in full words and appeared in the note below (i.e., SMS, short message service) the Table 1.

4. The abbreviation (i.e. TV) should be defined in full words and appeared in the note below (i.e., TV, Television) the Figure 1 and Figure 2.

5. Page 6, lines 155-156: “…(Table 2Error! Not a valid bookmark self-reference).” Please check and correct this.

6. In Table 2 and Table 3, a p value of 0.000 indicates that the values rounded to 0 but are not exactly 0. For example, a p-value like 0.0004 would be truncated to 0.000. Please revise P = 0.000 to P < 0.001.

Author Response

Reviewer 1

We thank the reviewer for their helpful comments and provide responses below:

Major:

  1. In the Materials and Methods section,

(1) page 14, line 316: the authors need to carefully state the survey period (e.q. ….between January 1, 2018 and March 31, 2019), not only the year.

Thanks, we have added this detail.

(2) respondents answered “don’t know” may imply that one lack knowledge or interest in the questions about awareness and knowledge of antibiotics and antibiotic resistance asked in the survey. In addition, it cannot be determined whether “don’t know” responses reflected an uncertainty which information had been seen or a desire to minimize response time. Please explain and add more information to a discussion.

Answering “don’t know” did not shorten response time compared to answering “no”, both answers were treated in the same way in the branching logic of the questionnaire. Therefore, we do not believe that respondents answered “don’t know” just to speed the interview up. We gave respondents the option to answer “don’t know” as well as “no” because of the limited understanding about different types of medicines in this community, and some respondents were not sure.

(3) Why did the authors group ‘no’ and ‘don't know’ together in the bivariable and multivariable regression models? Please explain and clarify more detail in the text.

We grouped “no” and “don’t know” together because we wanted to compare those that were confident that they had heard of antibiotics (i.e. those who answered “yes”) with those who were not confident about what antibiotics were (i.e those who had not heard of antibiotics – “no” – or were uncertain about what antibiotics were – “don’t know”). There were too few respondents who answered “don’t know” for this to be a category on its own, and we wanted to use a binary outcome variable for analysis. We have added text to the “Definition of variables” section (lines 348-353) to explain this.

  1. In the Results section,

(1) the authors have to show the aOR/aB, 95%CI and p-value of covariates for age, sex, highest level of education, occupation and household wealth tertile in multivariable logistic and linear regression models in Table 2 and Table 3.

What the reviewer describes would be appropriate for a predictive modelling approach. However, we have used a causal modelling approach, which provides an effect estimate for each explanatory variable adjusted for a small number of confounding variables (in our case age, sex, and education), rather than including all possible variables in one model. A discussion of the differences between predictive and causal models can be found here: https://statisticalhorizons.com/prediction-vs-causation-in-regression-analysis/  

(2) according to the results of Table 2 and Table 3, why did the authors not present the aB values of significant variables in the sentences? Please clarify more detail in the text.

In order to keep to the word limit, we did not include aB for linear regression models in the text, but explained that the results were qualitatively similar to the aORs, with a few exceptions.

  1. In the Discussion section,

(1) page 14, line 306‒310: a certain degree of recall bias is an inherent limitation to the study that uses self-reported survey data. In terms of attitudes and experiences of the household respondents, the varied reasons for awareness and knowledge of common antibiotics and antibiotic resistance and healthcare seeking, this could lead to the recall bias.

Thanks for this comment. We have added recall bias in this paragraph on limitations.

(2) other unmeasured factors, such as access to medical care, medication use, and the average number of medical outpatient visits in the past month may also influence the awareness and knowledge of antibiotics and healthcare seeking among the household respondents.

You are right, these factors are important. In terms of access to medical care, we already included distance from nearest health facility, usual health facility, and having medical insurance in our models, and did find some associations with antibiotic and AMR knowledge. We plan to explore the association between knowledge and antibiotic use in a future publication.

(3) these results may generalize only to a rural community in northern Vietnamese because of diverse household’s practice patterns. The results are less generalizable to other rural communities, provinces or countries. Please make clear about the limitation of the study in the text.

We have added a point about generalizability.

  1. In the Conclusion section, the description is too general and simplistic (i.e.,knowledge about types of antibiotics, awareness of antibiotic resistance, and determinants of antibiotic and antibiotic resistance awareness). These significant differences of main findings need to be summarized and rewritten. Please clarify and modify this more detail in the text.

Thanks for this suggestion. We have added more detail to the conclusion.

  1. The ‘citation formatting’ of several references (Ref. 1, 4, 7, 8, 12, 13, 15, 16, 18, 19, 22, 27, 29) is inaccuracy and inconsistent, for example the article title, journal name, page number, etc.. Please check reference formatting in detail according to the instructions of the Information for Authors of Antibiotics.

Thanks for pointing this out. We have corrected this.

Minor:

  1. Abbreviations (i.e., AMR, WHO, ….) should be defined in full words (i.e., antimicrobial resistance (AMR), World Health Organization (WHO), ….) in the first time of use, and then the abbreviation can be use many times.

We have corrected this.

  1. Page 2, lines 65: “…have a positive impact, although much of this evidence…” Please delete the comma after “impact”.

We have corrected this.

  1. The abbreviation (i.e. SMS) should be defined in full words and appeared in the note below (i.e., SMS, short message service) the Table 1.

We have corrected this.

  1. The abbreviation (i.e. TV) should be defined in full words and appeared in the note below (i.e., TV, Television) the Figure 1 and Figure 2.

We have corrected this.

  1. Page 6, lines 155-156: “…(Table 2Error! Not a valid bookmark self-reference).” Please check and correct this.

We have corrected this.

  1. In Table 2 and Table 3, a p value of 0.000 indicates that the values rounded to 0 but are not exactly 0. For example, a p-value like 0.0004 would be truncated to 0.000. Please revise P = 0.000 to P < 0.001.

We have corrected this.

Reviewer 2 Report

Godwin et al. have done a cross-sectional survey in the rural district of Ha Nam Province, Vietnam regarding awareness of antibiotics and antibiotic resistance. The study reports that most respondents (71.8%) had heard of antibiotics but only a few are aware of antibiotic resistance (17.7%). Also, antibiotic resistance awareness was lower among those who used private providers or pharmacies as their usual health facilities. The authors have concluded that to reach the highest communication among people use of multiple media channels and health information sources will be beneficial.

    It is not a surprise that the rural population has less knowledge about the AMR  however this kind of study should be done more so that health organizations will give more attention to people in those regions. The study methods and results were organized well.

  It will be attractive if the authors add a figure showing the flowchart of sample inclusion and exclusion criteria used in the present study.

Author Response

Reviewer 2

We thank the reviewer for their positive feedback.

It will be attractive if the authors add a figure showing the flowchart of sample inclusion and exclusion criteria used in the present study.

Thank you for this suggestion. The sample inclusion/exclusion process was quite simple and we didn’t think a flowchart was necessary, but we have now added one.

Round 2

Reviewer 1 Report

As for Comment #2(1), the authors replied that using a “causal modeling approach” to provide an effect estimate for each explanatory variable adjusted for a small number of confounding variables. Nevertheless, the study does not build a sufficiently causal model to link the levels of awareness and knowledge of antibiotics and antibiotic resistance. It is not reasonable to use causal language throughout the paper. Please check carefully.

In addition, in the limitation of the Discussion section, the authors said “we were unable to establish the direction of causality between awareness and knowledge of antibiotics and antibiotic resistance and healthcare seeking (lines 297-299) “ due to the cross-sectional nature of the study.

Otherwise, I have no further comment.